# Structure-based membrane dome mechanism for Piezo mechanosensitivity

**Yusong R Guo, Roderick MacKinnon***

Laboratory of Molecular Neurobiology and Biophysics, Howard Hughes Medical Institute, The Rockefeller University, New York, United States

**Abstract** Mechanosensitive ion channels convert external mechanical stimuli into electrochemical signals for critical processes including touch sensation, balance, and cardiovascular regulation. The best understood mechanosensitive channel, MscL, opens a wide pore, which accounts for mechanosensitive gating due to in-plane area expansion. Eukaryotic Piezo channels have a narrow pore and therefore must capture mechanical forces to control gating in another way. We present a cryo-EM structure of mouse Piezo1 in a closed conformation at 3.7Å-resolution. The channel is a triskelion with arms consisting of repeated arrays of 4-TM structural units surrounding a pore. Its shape deforms the membrane locally into a dome. We present a hypothesis in which the membrane deformation changes upon channel opening. Quantitatively, membrane tension will alter gating energetics in proportion to the change in projected area under the dome. This mechanism can account for highly sensitive mechanical gating in the setting of a narrow, cation-selective pore.
DOI: https://doi.org/10.7554/eLife.33660.001

## Introduction

Piezo ion channels, Piezo1 and Piezo2, are mechanosensitive channels (MS channels) that underlie force-detection in eukaryotic cells (*Coste et al., 2010*; *Wu et al., 2017*). The number of cellular processes found mediated by Piezo channels is large and growing at a rapid rate (*Eisenhoffer et al., 2012*; *McHugh et al., 2012*; *Pathak et al., 2014*; *Li et al., 2014*; *Cahalan et al., 2015*; *Alper, 2017*). Their essential biophysical characteristics include responsiveness in gating to mechanical force (i.e. mechanosensitivity) and selectivity of the pore for cations (*Gnanasambandam et al., 2015*; *Lewis and Grandl, 2015*). Together, these characteristics allow Piezo channels to serve their many biological functions, including the transduction of mechanical forces into electrical signals in sensory neurons (*Kim et al., 2012*; *Faucherre et al., 2013*; *Woo et al., 2014*).

Biophysical studies suggest that Piezo channels sense force directly through the lipid membrane (*Cox et al., 2016*; *Syeda et al., 2016*), although other mechanisms also have been proposed (*Lewis and Grandl, 2015*; *Gottlieb et al., 2012*; *Peyronnet et al., 2013*; *Poole et al., 2014*; *Borbiro et al., 2015*). To understand how mechanical forces might influence the balance between opened and closed states in Piezo channels it is useful to consider the most thoroughly studied and best-understood MS channel, MscL from bacteria (*Sukharev et al., 1997*; *Perozo et al., 2002*; *Moe and Blount, 2005*; *Li et al., 2009*; *Iscla and Blount, 2012*). Structural, biophysical and functional analyses show that MscL undergoes a large (~20 nm$^2$) in-plane area expansion when it opens (*Perozo et al., 2002*; *Chang et al., 1998*; *Sukharev et al., 2001*; *Corry et al., 2010*). This area expansion produces a decrease in the free energy of the membrane-channel system if the membrane is under tension, favoring the open state (*Wiggins and Phillips, 2005*). An illustration adapted from *Ursell et al. (2008)* provides a 'gravitational analog' for understanding the origin of the free energy change: as the channel opens and expands in the plane of the membrane, weights are lowered (*Figure 1*). The free energy difference between closed and opened conformations has been described as

*For correspondence: mackinn@rockefeller.edu

**Competing interests:** The authors declare that no competing interests exist.

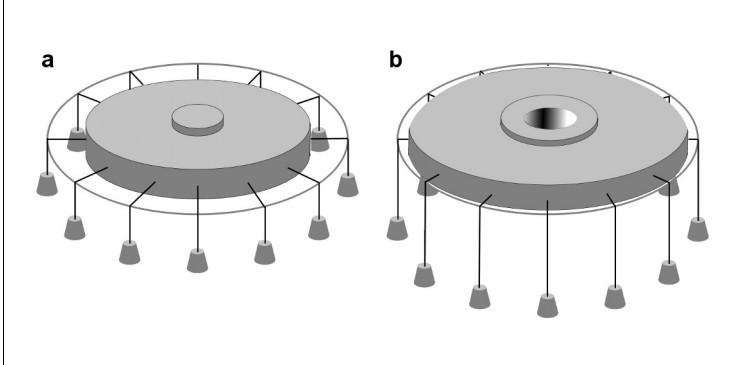

**Figure 1.** In-plane area expansion lowers the free energy of a channel-membrane system under tension. Analog of a membrane under tension showing tethered weights in a gravitational field pulling the membrane taut, adapted from *Ursell et al. (2008)*. (a) When the channel is closed, potential energy from the weights is high. (b) When the channel opens and in-plane area expands, the weights are lowered, and the potential energy of the system is decreased.

DOI: https://doi.org/10.7554/eLife.33660.002

$$\Delta G = \Delta G(\gamma = 0) - \gamma \Delta A \tag{1}$$

where $\Delta G(\gamma = 0)$ is the free energy difference at zero tension, $\gamma$ the membrane tension and $\Delta A$ the in-plane cross sectional area change associated with channel opening (*Haswell et al., 2011*). Thus, open state stabilization relative to closed is proportional to membrane tension and the proportionality constant is in-plane area expansion. That is, the sensitivity of gating with respect to membrane tension is

$$d(\Delta G)/d(\gamma) = -\Delta A. \tag{2}$$

In MscL the 20 nm$^2$ physical expansion determined by structural and biophysical analysis (*Perozo et al., 2002*; *Sukharev et al., 2001*; *Corry et al., 2010*) corresponds well to the tension dependence of gating estimated in the most carefully executed functional experiments (*Chiang et al., 2004*).

The principle of mechanosensitivity exhibited by MscL suggests a simple recipe for mechanosensitivity in general: couple pore opening to in-plane area expansion. But as a general solution this principle seems problematic for Piezo channels. It works well for MscL because that channel functions as a 'pressure release valve', permitting bacterial survival in the face of osmotic shock (*Sukharev et al., 1993*). The wide pore opening in MscL, associated with a conductance of approximately 3 nS and complete lack of ion selectivity, naturally fits with a large in-plane area expansion and therefore high mechanosensitivity (*Sukharev et al., 1993*). By contrast Piezo channels have conductance values around 30 pS (100 times smaller than MscL) and are cation selective, characteristics that are incompatible with a wide pore (*Coste et al., 2010*). Yet Piezo channels are highly mechanosensitive (*Lewis and Grandl, 2015*; *Cox et al., 2016*). Somehow nature must have separated area expansion from pore diameter. In an attempt to understand how this was accomplished we have determined a structure of the mouse Piezo1 (mPiezo1) channel, which exhibits many differences in its molecular detail and global interpretation when compared to an earlier structure (*Ge et al., 2015*). Based on this new structure, we propose a mechanism for understanding Piezo channel mechanosensitivity.

## Results

### Structure of mPiezo1

Using cryo-electron microscopy (cryo-EM) we determined a structure of mPiezo1 to an overall resolution of 3.7 Å (*Figure 2* and *Figure 2—figure supplement 1*). Viewed down its 3-fold axis, the

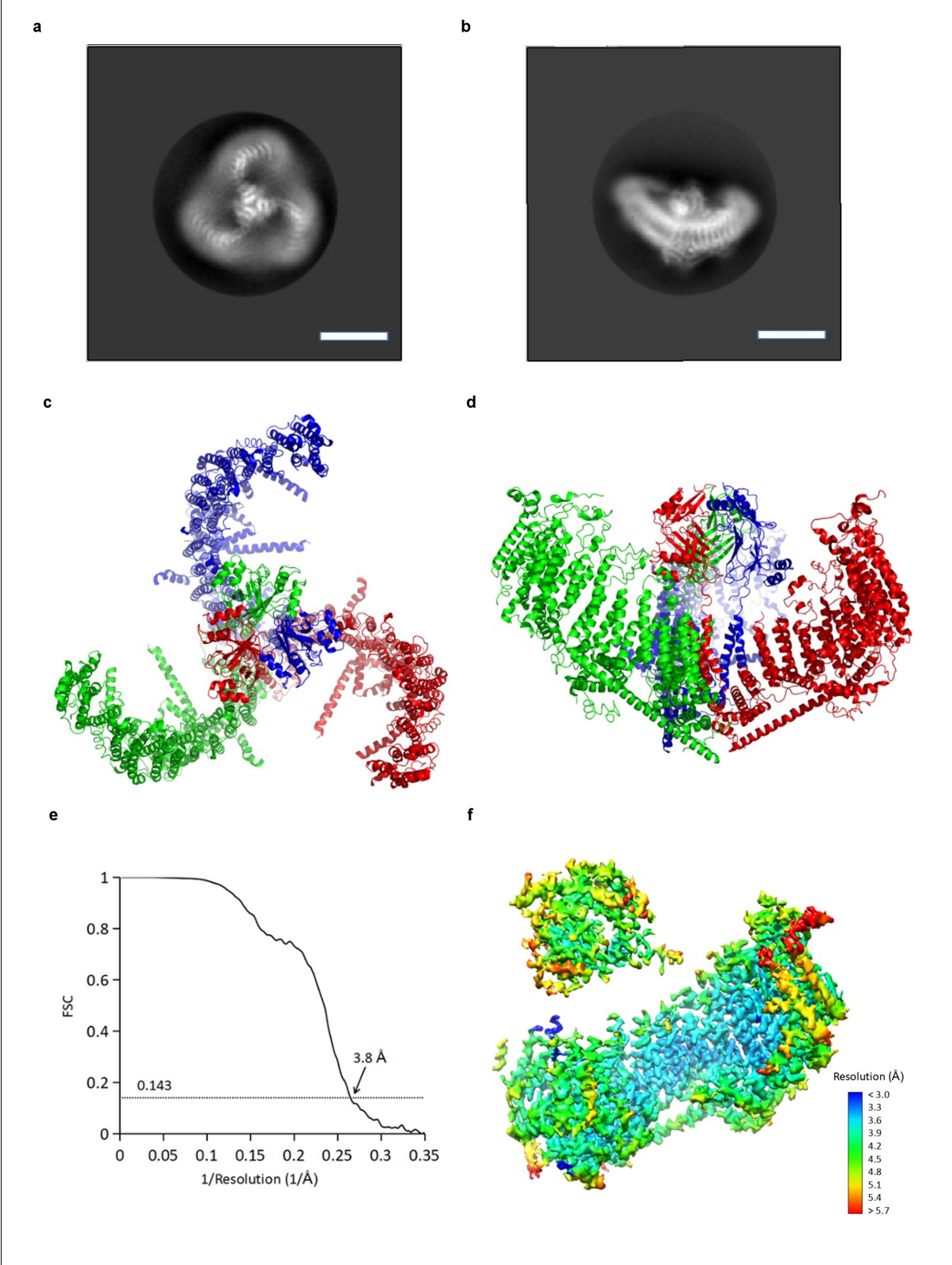

**Figure 2.** CryoEM reconstruction of mPiezo1. (**a and b**) Representative 2D averaged classes, viewed from the top (**a**), and the side (**b**), scale bar 10 nm. (**c and d**) Atomic model of the trimeric channel shown as ribbon diagram, viewed from the top (**c**), and the side (**d**). The three subunits are colored in red, green and blue, respectively. (**e**) Fourier shell correlation (FSC) curves calculated between two half maps after C1 masked refinement and post-

*Figure 2 continued on next page*

*Figure 2 continued*

processing in RELION. (**f**) Local resolution of density map from C1 masked refinement, estimated by Blocres. The map shown is low-pass filtered to 3.8 Å and sharpened with a b-factor of −200 Å$^2$.

DOI: https://doi.org/10.7554/eLife.33660.003

The following figure supplements are available for figure 2:

**Figure supplement 1.** CryoEM structure determination of mPiezo1.
DOI: https://doi.org/10.7554/eLife.33660.004
**Figure supplement 2.** Local EM densities of mPiezo1 (Residues 581–1231).
DOI: https://doi.org/10.7554/eLife.33660.005
**Figure supplement 3.** Local EM densities of mPiezo1 (Residues 1280–2543).
DOI: https://doi.org/10.7554/eLife.33660.006

trimeric Piezo channel is a triskelion with extended arms (*Figure 2a,c*); from the side, it is curved as shown (*Figure 2b,d*). The cryo-EM map shows greatest detail near the 3-fold center (local resolution 3.2 Å to 3.6 Å) and less detail near the periphery (local resolution 5 Å to 6 Å) (*Figure 2—figure supplement 1e*). To improve the map quality, symmetry expansion was conducted with focused refinement using a mask covering the central hub and one of the three extended arms. The resulting map showed improvement, especially at the periphery (local resolution 4 Å to 5 Å) (*Figure 2f* and *Table 1*). A crystal structure (PDB ID: 4RAX) was docked into density corresponding to the C-terminal extracellular domain (CED) and the remaining model (26 TM helices per subunit) was built *de novo* (*Figure 2—figure supplements 2* and *3*) (*Ge et al., 2015*). The refined structural model contains 1518 residues (out of 2547), with the N-terminal 576 residues and internal flexible loops missing.

The structure's extended arms consist of 24 helices arranged as six repeated 4-TM structural units (*Figure 3a,b*). Each unit is a left-handed bundle of four helices starting and ending on the intracellular side. The two extracellular loops (between the first and second, and third and fourth TM helices) are short and long, respectively, the latter being partially ordered and containing α-helices. The six 4-TM repeats assemble with each other also as a left-handed helix, causing the extended arms of the triskelion to spiral away from the trimer's center, out of plane with respect to the central pore axis (*Figures 2c,d* and *3b*). Adjacent 4-TM repeats are linked by a polypeptide chain that crosses the span of two repeats, owing to their relative orientations (*Figure 3b*). The interactions provided by an extended connection likely aid stabilization of the extended arms. The repetitive pattern of TM helices is well conserved for all six 4-TM units visible in the structure. Sequence analysis suggests that the pattern continues to the N-terminus (*Figure 3—figure supplement 1–3*). Therefore, it is likely that a Piezo subunit contains nine 4-TM units altogether (the first 3 of which we do not see) giving a total of 36 TM helices, which form the arms, plus two C-terminal TM helices that meet at the center of the trimer. We have numbered helices in our structural model accordingly (*Figure 3a*).

The central hub of the trimer is formed by amino acids following TM36 (*Figure 3* and *Figure 6—figure supplement 1*). At the center TM37-38 form a 3-fold symmetric channel lined by TM38, capped on the extracellular side by a CED, and extended on the intracellular side by a pore extension (PE) helix (*Figure 3a* and *Figure 6—figure supplement 1c,d*). TM37-38 are domain-swapped relative to TM1-36 (*Figure 2c*). The channel is surrounded at the level of the inner membrane leaflet by a layered, helical cuff. The cuff consists of 'elbow' helices (residues 2116 to 2142), a 'base' helix (residues 2149 to 2175) and 'hairpin' helices (residues 2501 to 2534) from all three subunits (*Figure 3a* and *Figure 6—figure supplement 1c,d*). This solid cuff attaches the extended arms to the channel.

Piezo also contains a long intracellular helix of 66 amino acids extending from the trimer center, radially out to TM28 (*Figures 2d* and *3*). Referred to as a 'beam', this feature was observed previously (*Ge et al., 2015*), but modeled as two instead of one helix. In addition, linkers between 4-TM units contain at least one helix that runs perpendicular to the TM helices and to the extended arms (*Figures 2c*, *3* and *7b*). These 'cross' helices are mostly hydrophobic and located inside of the micelle density, near the intracellular interface.

**Table 1.** Cryo-EM data collection, refinement and validation statistics.

| | mouse Piezo1 (EMDB-7042) (PDB 6B3R) |
|---|---|
| **Data collection and processing** | |
| Magnification | 22,500 |
| Voltage (kV) | 300 |
| Electron exposure (e⁻/Å²) | 47 |
| Defocus range (μm) | 1.0–2.4 |
| Pixel size (Å) | 1.3 |
| Symmetry imposed | C1 |
| Initial particle images (no.) | 1 |
| Final particle images (no.) | 50 |
| Map resolution (Å)<br>    FSC threshold | 3.8<br>0.143 |
| Map resolution range (Å) | 3.2–10.7 |
| **Refinement** | |
| Initial model used (PDB code) | 4RAX |
| Map sharpening $B$ factor (Å²) | −200 |
| Model composition<br>    Non-hydrogen atoms<br>    Protein residues<br>    Ligands | <br>35730<br>4554<br>0 |
| $B$ factors (Å²)<br>    Protein<br>    Ligand | <br>302.4<br>N/A |
| R.m.s. deviations<br>    Bond lengths (Å)<br>    Bond angles (°) | <br>0.006<br>1.032 |
| Validation<br>    MolProbity score<br>    Clashscore<br>    Poor rotamers (%) | <br>1.73<br>4.84<br>0.25 |
| Ramachandran plot<br>    Favored (%)<br>    Allowed (%)<br>    Disallowed (%) | <br>91.96<br>8.04<br>0 |

DOI: https://doi.org/10.7554/eLife.33660.007

## Membrane-curving properties of Piezo

A single subunit of Piezo removed from the trimer can be positioned reasonably well into the plane of a lipid membrane (*Figure 4a*). However, the detergent micelle containing a trimer is curved into a dome shape (*Figures 2b,4b*). The diameter of the dome opening is about 18 nm with a depth of about 6 nm. The central cap, formed by the CED, is actually located mostly inside the dome. The angle between the central pore axis and the beam in each subunit is about 60° instead of 90°, as we would expect if the trimer were located in a fully flattened membrane (*Figures 2d,4b, c*). Consequently, the hydrophobic residues on the TM helices (flanked by charged amino acids) form a clearly curved band on the trimer surface, matching the micelle density (*Figure 4c*). The distribution of charged amino acids on the extended arms of Piezo is consistent with the positive inside rule for membrane proteins (*von Heijne, 1992*).

The unusual shape of the Piezo trimer in detergent micelles led us next to test whether such a non-planar conformation can occur inside a lipid bilayer. *Figure 5a* shows a Piezo channel reconstituted into a small unilamellar vesicle consisting of POPE and POPG. The presence of a Piezo channel produces a local dome-shaped deformation of the membrane, with density corresponding to the CED visible inside the dome (the channel is inserted with its extracellular side inside the vesicle). The

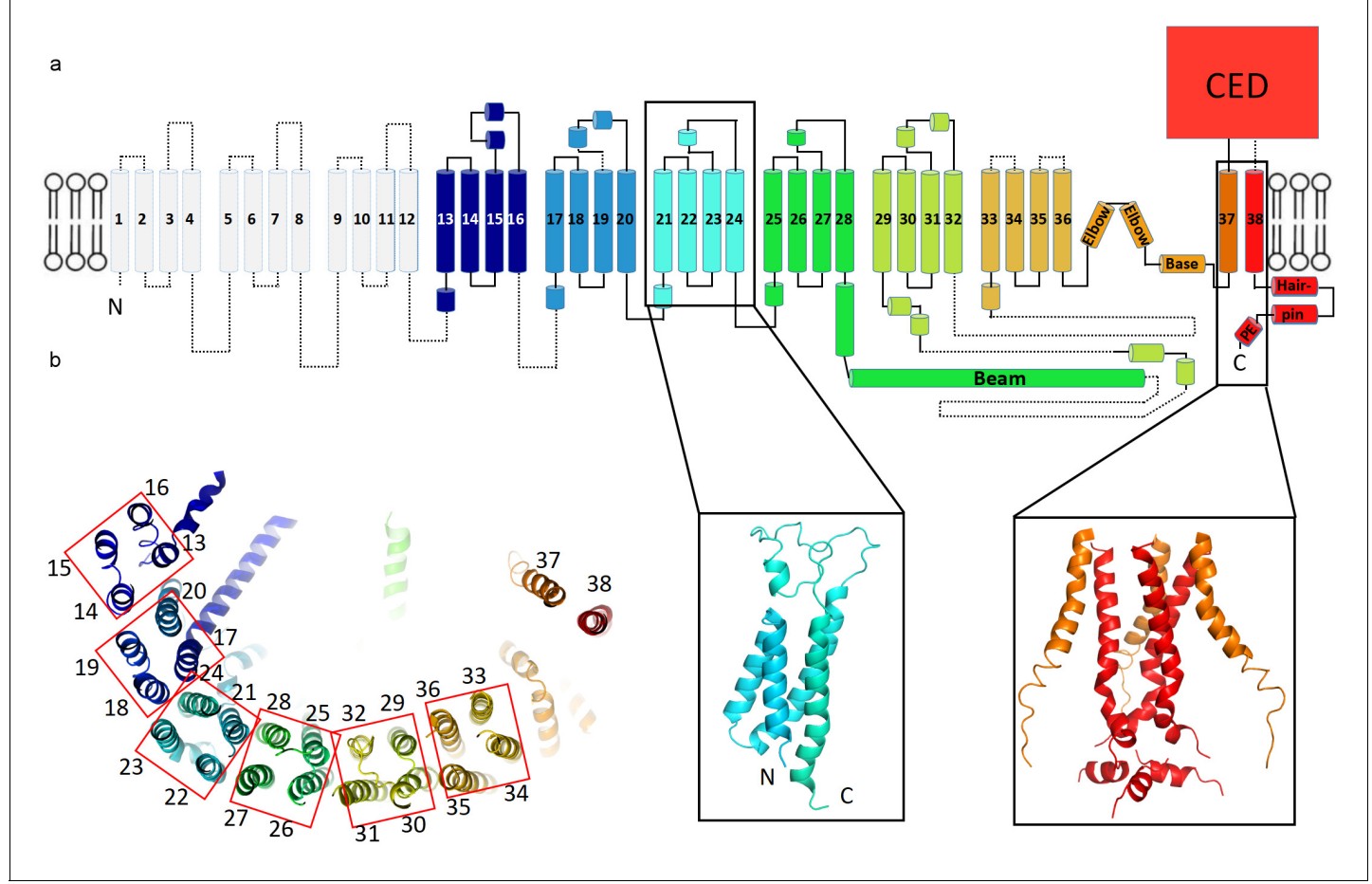

**Figure 3.** Topology of mPiezo1. (a) Cartoon representation of a monomer, rainbow-colored with C-terminus in red and N-terminus in blue, except for the first 12 TMs that are not visible in our structure. Helices within a single 4-TM unit are colored uniquely. Helices are shown as cylinders, loops as solid lines, and unresolved regions as dotted lines. C-terminal extracellular domain (CED) is simplified as a box. A ribbon diagram of 4-TM unit 6, consisting of TM 21 to 24, is shown in the left inset panel with N- and C- termini labeled. The right inset panel shows a ribbon diagram of the pore region, formed by TM37, TM38 and the PE helix from all three subunits. (b) A ribbon diagram of a monomer rainbow-colored as in A, viewed from top. Each 4-TM unit is highlighted in a red box with TM number labeled.

DOI: https://doi.org/10.7554/eLife.33660.008

The following figure supplements are available for figure 3:

**Figure supplement 1.** Hydrophathy analysis of mPiezo1 (Residues 1–900).
DOI: https://doi.org/10.7554/eLife.33660.009
**Figure supplement 2.** Hydrophathy analysis of mPiezo1 (Residues 901–1800).
DOI: https://doi.org/10.7554/eLife.33660.010
**Figure supplement 3.** Hydrophathy analysis of mPiezo1 (Residues 1801–2547).
DOI: https://doi.org/10.7554/eLife.33660.011

molecular model, without adjustment, conforms to the shape of the locally curved membrane near the Piezo channel (*Figure 5a*).

We observed that vesicles consisting of POPE and POPG tended to be non-spherical even in the absence of Piezo channels, possibly due to the truncated cone and inverted, truncated cone shapes of POPG and POPE lipids, respectively (*Israelachvili, 1992*). To further examine the ability of Piezo to deform membrane bilayers, we also analyzed vesicles consisting of mixtures of POPC and DOPS, with and without cholesterol. Vesicles of these lipid compositions are spherical in shape (circular in projection) (*Figure 5b*). As in POPE and POPG vesicles, Piezo channels deform the membrane by producing local membrane curvature (*Figure 5c* and *Figure 5—figure supplement 1*). We note that Piezo channels are almost always inserted with an inside-out orientation in vesicles so that the local

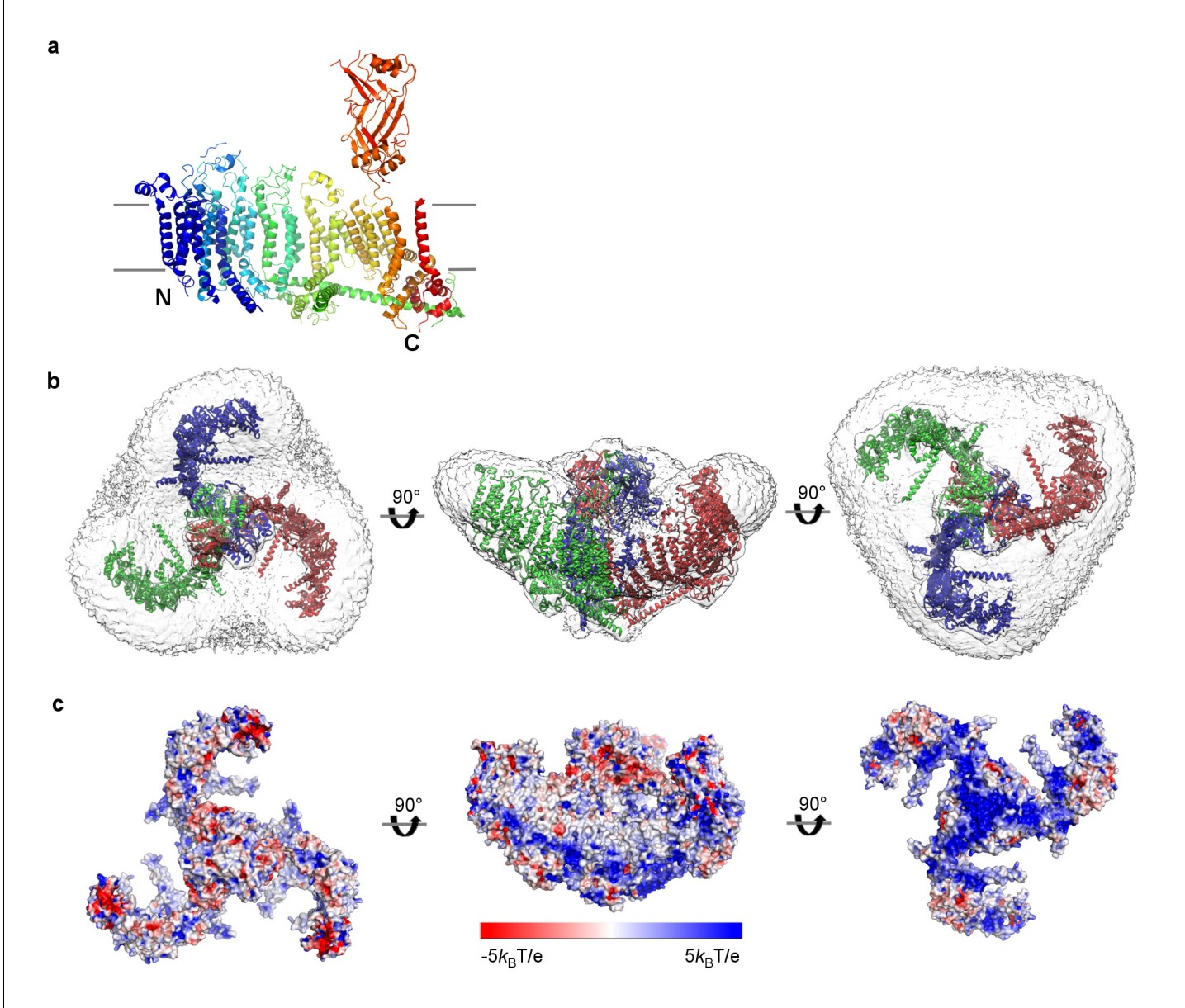

**Figure 4.** mPiezo1 trimer in curved micelle. (**a**) The same ribbon diagram of a monomer taken from the trimer, as in 3B, viewed from the side, with N- and C- termini labeled. Approximate locations of planar membrane interfaces are shown as grey lines. (**b**) Ribbon diagrams of a trimer in an unsharpened map, contoured at 6σ, showing micelle density. Top, side and bottom views are shown. (**c**) Surface representation of a trimer, colored based on electrostatic potentials in aqueous solution containing 150 mM NaCl, calculated using APBS, with positive shown blue, neutral white, and negative red. Top, side and bottom views are shown.

DOI: https://doi.org/10.7554/eLife.33660.012

The following figure supplement is available for figure 4:

**Figure supplement 1.** Interface between CED and TM loops.

DOI: https://doi.org/10.7554/eLife.33660.013

curvature (of the dome) matches the global curvature of the vesicle. We conclude that Piezo deforms lipid bilayers locally into a dome shape. In cells, the dome will project towards the cytoplasm.

A negatively charged patch of amino acids (E2257, E2258 and D2264) on the CED lies in close contact with a positively charged patch (R1761, R1762 and R1269) located in the extracellular loops of 4-TM units 2 and 3 (*Figure 4—figure supplement 1*). Hydrogen bonds and salt bridges connect

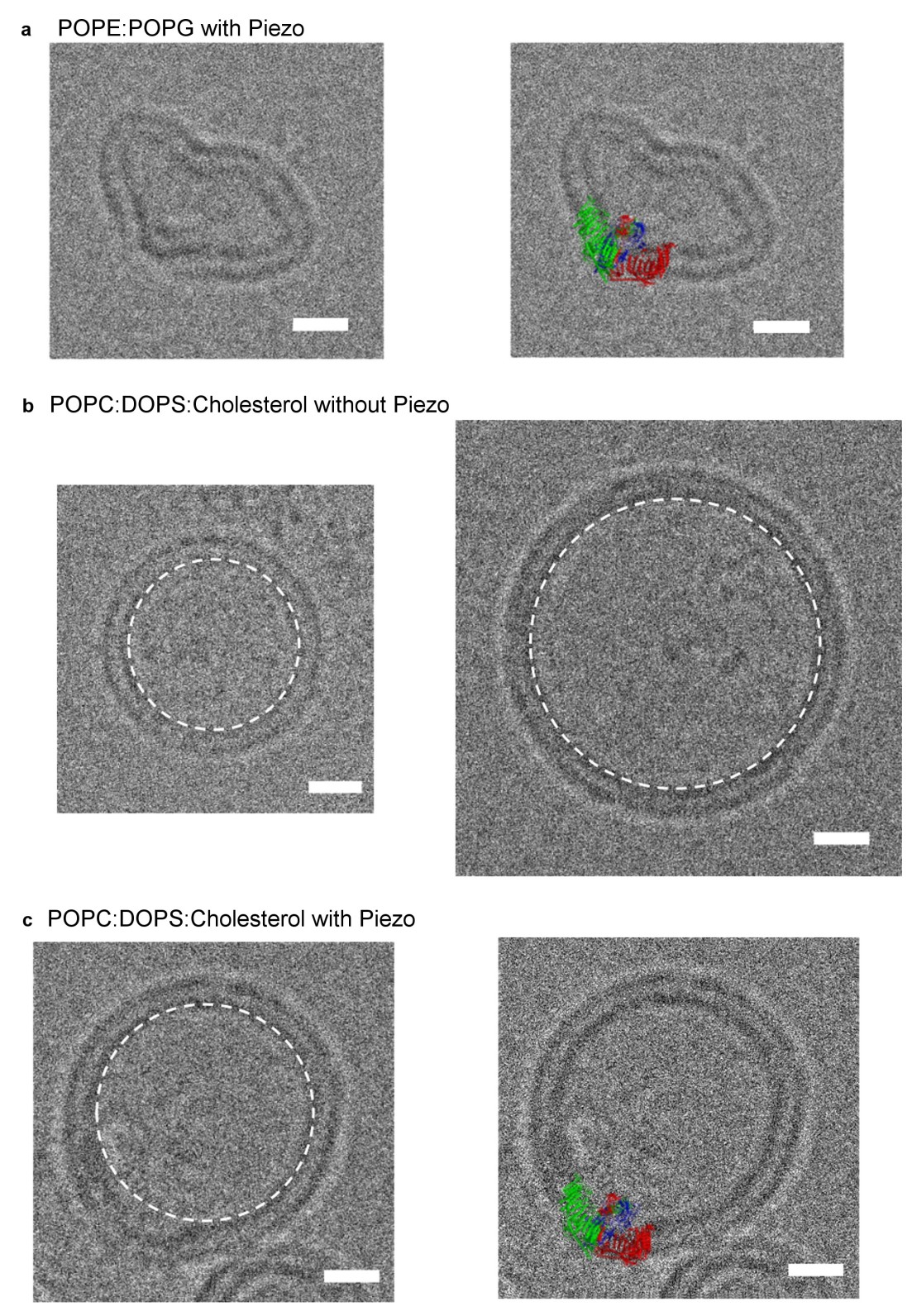

**Figure 5.** mPiezo1 trimer locally curves membrane. (a) Small unilamellar vesicle containing Piezo without (left) and with (right) molecular model scaled to size and inserted into image. The lipid composition is POPE:POPG = 3:1 wt ratio. Scale bar 10 nm. (b) Small unilamellar vesicles containing no protein. The lipid composition is POPC:DOPS:cholesterol = 8:1:1 wt ratio. The spherical vesicle projection is highlighted by a white dashed circle. Scale

*Figure 5 continued*
bar 10 nm. (**c**) Small unilamellar vesicle containing Piezo without (left) and with (right) molecular model scaled to size and inserted into image. The lipid composition is POPC:DOPS:cholesterol = 8:1:1 wt ratio. The spherical vesicle projection is highlighted by a white dashed circle. Scale bar 10 nm.
DOI: https://doi.org/10.7554/eLife.33660.014
The following figure supplement is available for figure 5:

**Figure supplement 1.** mPiezo1 trimer in liposomes of various sizes.
DOI: https://doi.org/10.7554/eLife.33660.015

E2257 to R1762 and D2264 in R1761 in a domain-swapped manner. These electrostatic interactions appear to stabilize the trimeric assembly in its curved conformation.

## Structural features and conformational state of the pore

The transmembrane pore, lined by TM38 and the PE helix, is closed (*Figure 6a–c*). At the level of the membrane inner leaflet, the pore radius at positions E2537, P2536 and M2493 is 0.1 Å, 0.4 Å and 0.3 Å, respectively. Estimated from the size of tetraethylammonium (TEA), which is permeable, Piezo should open to a radius of at least 4 Å (*Gnanasambandam et al., 2015*). When it does open, two rings of glutamate residues flanking either side of the narrow region of the pore (E2487 and E2537) likely account for cation selectivity through direct interactions with permeating ions (*Figure 6a,c*).

The overall architecture of Piezo's pore shares several similarities with P2X (*Kawate et al., 2009*) and acid-sensing ion channels (ASIC) (*Baconguis et al., 2014*) (*Figure 6—figure supplement 1a,b*). First, all three are trimeric. Second, the pores contain two TM helices and a large extracellular domain, which enclose a continuous ion-conduction path along the three-fold symmetry axis, including a central vestibule in the extracellular domain. Third, there are lateral fenestrations located

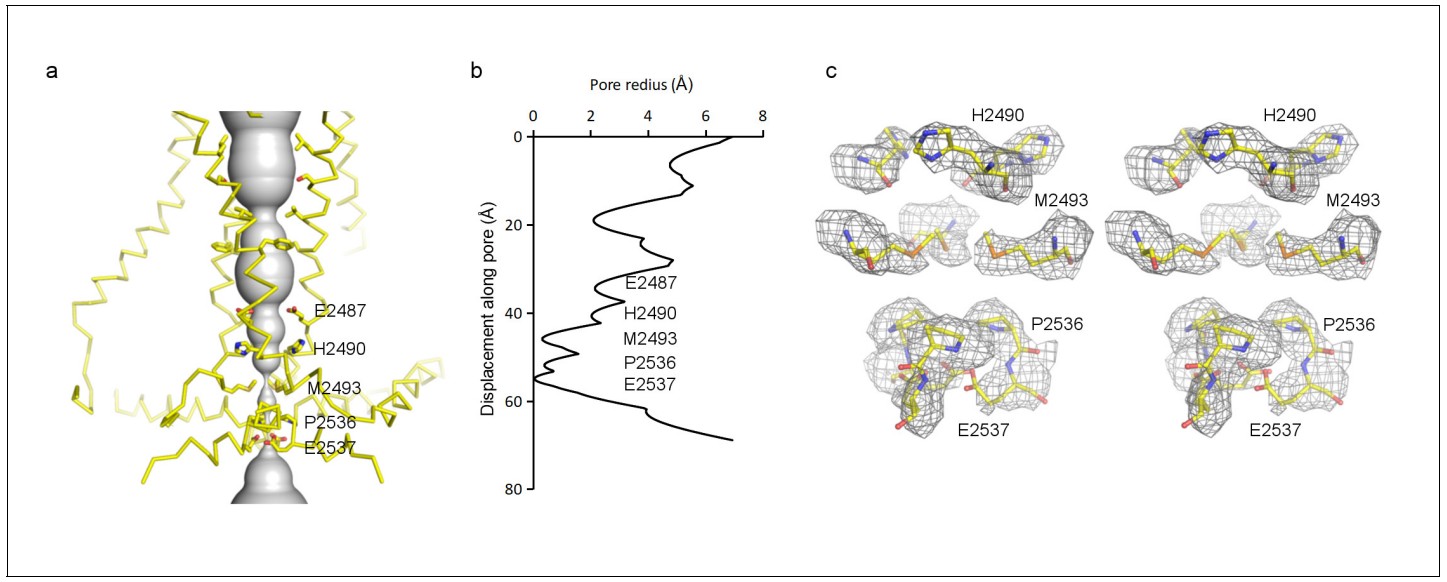

**Figure 6.** Pore of the mPiezo1 channel. (**a**) Ion-conduction path viewed from the side. The distance from the pore axis to the protein surface is shown as grey sphere. Cα trace of the pore (TM37, TM38, hairpin and PE helices) is shown in yellow. Residues facing the pore are shown as sticks. Constricting residues are labeled. (**b**) Radius of the pore. The van der Waals radius is plotted against the distance from the top along the pore axis. Constricting residues are labeled as in A. (**c**) Stereo view of the side-chain density around constricting residues. The map is contoured at 6σ and sharpened with a b-factor of −200 Å². The atomic model is shown as sticks, colored according to atom type: yellow, carbon; red, oxygen; blue, nitrogen; and orange, sulfur.
DOI: https://doi.org/10.7554/eLife.33660.016
The following figure supplement is available for figure 6:

**Figure supplement 1.** Pore region of the mPiezo1 channel.
DOI: https://doi.org/10.7554/eLife.33660.017

below the extracellular domain and open to the pore, which is the actual ion-conduction path in P2X and ASIC (*Kellenberger and Grutter, 2015*), and could be the case for Piezo as well. Given their distinct genetic backgrounds yet similar overall structures, it is reasonable to hypothesize that the two TM helices with an extracellular domain are the core components for a functional and efficient trimeric channel of this type. Therefore, other components unique to Piezo, including the elbow, base, hairpin, and PE helices (*Figure 3a* and *Figure 6—figure supplement 1c,d*), are likely to be directly associated with mechano-transduction. These unique components are positioned in a cuff surrounding the narrowest region of the pore and therefore seem well positioned to influence gating. Upon mechanical stimulus, major conformational changes are expected in these components in response to movement of the triskelion arms to open the pore.

## Discussion

### Hypothesis for mechanical gating

Piezo has a central ion conduction pathway surrounded by extended arms composed of transmembrane helices. The most strikingly unusual feature of Piezo's structure is that it does not conform to a locally planar membrane. Instead, its extended arms project approximately 30° out of the plane defined by the pore, incompatible with a flat membrane (*Figures 2d,4b,c*). The extended arms are also curved so that the entire structure takes the appearance of a 3-sided pyramid – inverted when viewed from outside the cell – that spirals from base to apex to form a semi-sphere-shaped dome (projecting into the cell). Hydrophobic helices lie perpendicular to the arms like cross bars between the arms, presumably to support the curved membrane as it conforms to Piezo's non-planar shape (*Figures 2,4,7b*). Twelve N-terminal transmembrane helices are not resolved. These presumably extend the arms even further, however, given the curvature of the arms these N-terminal helices appear to coincide with the edge of the dome, far from the center, where the membrane becomes planar again. We note that a published low-resolution structure led the authors to conclude that Piezo resides in a locally planar membrane, with extended arms forming extra-membranous, extracellular blades (*Ge et al., 2015*). The structure we have described is fundamentally different in that the arms, composed of transmembrane helices, are inside the membrane and force the membrane to curve, as shown in images of Piezo reconstituted into lipid vesicles (*Figure 5* and *Figure 5—figure supplement 1*).

We have approximated the shape of the membrane surrounding Piezo as a dome with a spherical surface of radius 10.2 nm, corresponding to the mid-plane surface of a membrane 3.6 nm thick (*Figure 7a,b* and *Figure 7—figure supplement 1a*). The shape matches closely but not perfectly the hydrophobic boundaries of Piezo, but it is sufficient for the following discussion. By appealing to the illustration of mid-plane area expansion in *Figure 1*, the structure of Piezo offers a plausible explanation for the origin of its tension-gating. If the semi-spherical dome becomes flatter (i.e. more co-planar with the membrane) when Piezo opens, then the channel-membrane system will expand. In MscL, widening of the channel's diameter causes membrane plane expansion. In the proposed mechanism for Piezo, flattening of the channel and its locally surrounding membrane causes membrane plane expansion by transferring out-of-plane membrane area (the dome) into the membrane plane. In both cases weights are lowered in the 'gravitational analog' (i.e. free energy is reduced) (*Figure 1*). Because flattening does not constrain the pore to open wide, expansion and pore diameter are decoupled such that Piezo can exhibit its small conductance and cation selectivity, properties that are essential to its function, without compromising the ability to sense lateral membrane tension.

In this proposed mechanism for Piezo, expansion of the membrane-channel system occurs when a curved, dome-shaped membrane becomes flatter (more planar). The expanded plane area corresponds quantitatively to the change in area of the dome that is projected onto the membrane plane, $\Delta A_{proj}$, and the resulting energy difference is $\gamma \Delta A_{proj}$ (*Figure 7c*). For the simplified geometry shown in *Figure 7* the semi-spherical mid-plane has a total mid-plane surface area of 400 nm$^2$ and a projected area, $A_{proj}$, of 280 nm$^2$ (*Figure 7—figure supplement 1a*). Therefore, if Piezo becomes completely co-planar with the membrane when it opens (i.e. so that its projected area equals its total area) then $\Delta A_{proj}$ associated with opening would be 400 nm$^2$ – 280 nm$^2$ = 120 nm$^2$. A membrane under tension $\gamma$ would thus favor the open conformation by energy $\gamma \Delta A_{proj}$, corresponding to 42

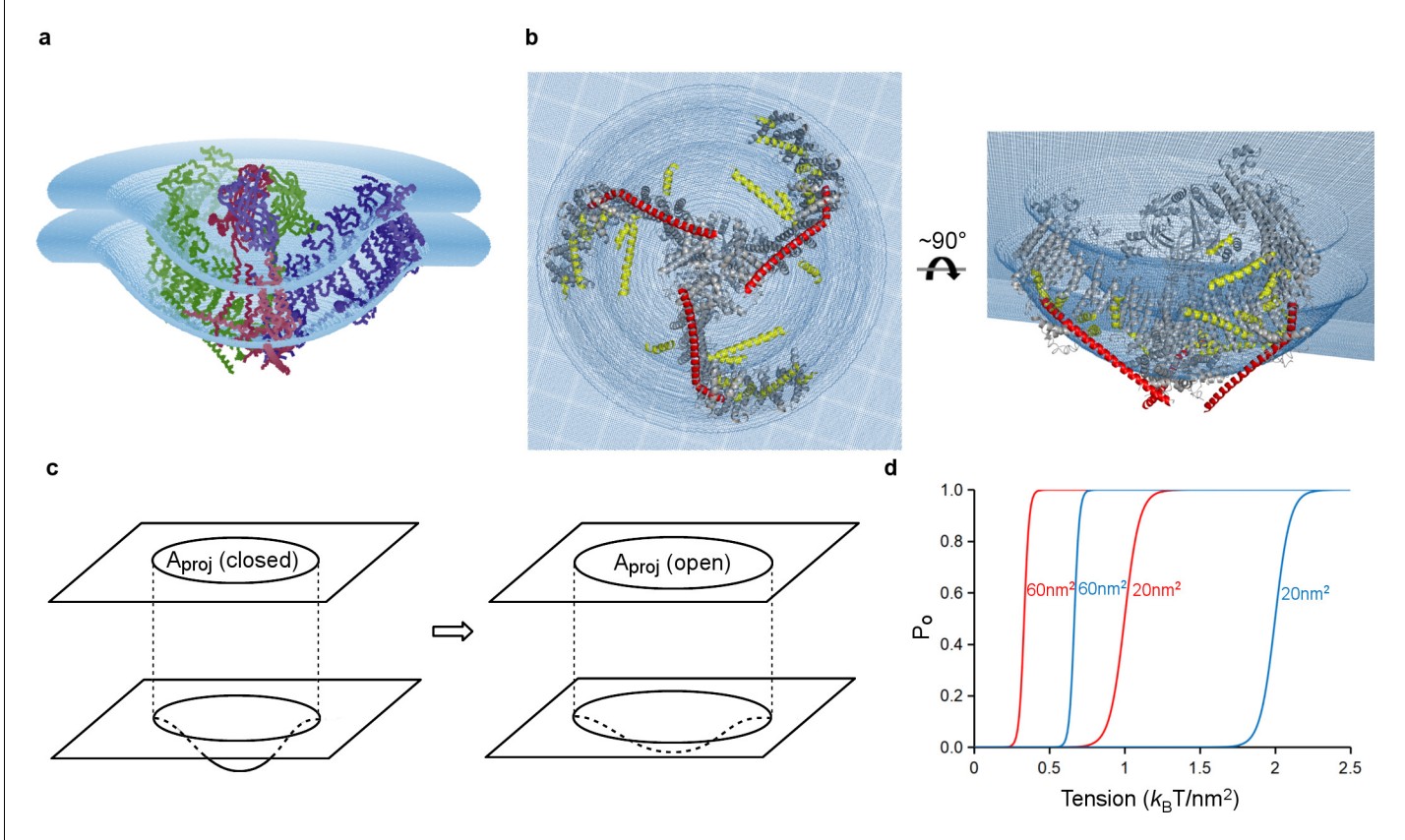

**Figure 7.** Model of tension-gating in mPiezo1. (**a**) Cα trace representation of a trimer placed in a semi-sphere-shaped membrane 3.6 nm thick, idealized from the curved micelle density. The mid-plane semi-sphere has radius of 10.2 nm and is centered 4.0 nm above the projection plane. The three subunits are shown in red, green and blue, respectively. (**b**) Ribbon diagram of a trimer in the idealized membrane. The 'beam' formed by residues 1300–1365 is highlighted in red, the cross-helices are highlighted in yellow, while the remaining protein is colored in grey. (**c**) Illustration of projection area (circle in top plane) changing as the surface curvature of the channel and local membrane (bottom plane) changes. (**d**) Theoretical activation curves corresponding $(\Delta G_{prot} + \Delta G_{bend})$=20 $k_B$T (red) or 40 $k_B$T (blue) and $\Delta A_{proj}$ = 20 nm$^2$ or 60 nm$^2$. The curves are generated through $P_o$ = $(1 + Exp[(\Delta G_{prot} + \Delta G_{bend}) - \gamma \Delta A_{proj}])^{-1}$.

DOI: https://doi.org/10.7554/eLife.33660.018

The following figure supplement is available for figure 7:

**Figure supplement 1.** References for area and energy calculations.
DOI: https://doi.org/10.7554/eLife.33660.019

$k_B$T stabilization of the open state relative to closed at a membrane tension of only one tenth lytic value (~3.5 $k_B$T/nm$^2$), where $k_B$ is the Boltzmann constant and T the temperature in degrees Kelvin (**Rawicz et al., 2000**). If Piezo flattens only partially upon opening then $\Delta A_{proj}$ will be less than 120 nm$^2$ and the energy difference correspondingly smaller (**Figure 7c,d**). Recall that for MscL, the in-plane area change $\Delta A$ is ~20 nm$^2$. This means Piezo has the potential for much higher tension sensitivity than MscL (**equation 2**, with $\Delta A = \Delta A_{proj}$ for Piezo). There are other possible origins of tension dependence in Piezo, related to changes in the size, geometry and chemistry of the protein-lipid interface associated with gating (**Wiggins and Phillips, 2005**; **Ursell et al., 2008**; **Phillips et al., 2012**). While these changes are still unknown, we suspect that the major origin of tension-dependent gating is the change in projected area, because it has the potential to be so large.

In addition to the tension-dependent energy associated with projected area expansion, there is a separate, intrinsic energy cost to bend a membrane even in the absence of tension (**Ursell et al., 2008**; **Helfrich, 1973**; **Deserno, 2007**; **Phillips, 2017**). For the semi-spherical membrane in **Figure 7** we calculate the magnitude of this energy to be approximately 150 $k_B$T, even excluding the curvature back into the membrane plane, which will substantially increase the bending energy further

(*Figure 7—figure supplement 1b*). This bending energy implies that the Piezo protein must perform work on the membrane to curve it. We propose that in the absence of a membrane's tendency toward planarity (i.e. in a detergent micelle) Piezo is very stable in its closed, curved conformation and we observe physical features – such as surface electrostatic interactions – that support this idea (*Figure 4—figure supplement 1*). Inside a membrane Piezo produces similar membrane curvature, but this must occur at the energetic expense of its closed state stability, because some of that energy must go into curving the membrane. Then, we propose, when tension is applied to the membrane, the additional energy term $\gamma \Delta A_{proj}$ tips the balance, favoring the flatter, open conformation (*Figure 7d*). A modification of the free energy *equation 1* for Piezo could be written

$$\Delta G = (\Delta G_{prot} + \Delta G_{bend}) - \gamma \Delta A_{proj}. \qquad (3)$$

The terms $\Delta G_{prot}$ and $\Delta G_{bend}$ refer to free energy differences intrinsic to protein gating and membrane bending, respectively. In this description, for the transition from closed to open, $\Delta G_{prot}$ is positive (unfavorable) and $\Delta G_{bend}$ is negative (favorable): these tend to cancel each other, with $\Delta G_{prot}$ somewhat greater in magnitude so that the channel is closed when $\gamma = 0$. The equation predicts that lipid membranes of differing stiffness (i.e. different bending moduli) should influence gating through $\Delta G_{bend}$ (i.e. stiffer membranes with a more negative $\Delta G_{bend}$ should lead to channel opening at smaller values of membrane tension) (*Figure 7d*). That $\Delta G_{bend}$ and $\Delta G_{prot}$ are indeed tension-independent (such that $d(\Delta G)/d\gamma = -\Delta A_{proj}$) is an assumption that needs to be tested through further measurement and analysis.

We can only conjecture how changes in Piezo's shape will open the pore. We think it is significant that a single subunit is compatible with a planar membrane, whereas the trimer is not. This is due to the abrupt angle at which the arms project from the pore (*Figures 2d*,*4b,c*): this implies that forces tending to flatten Piezo will produce the greatest stress at the region of attachment. This region coincides with the cuff of helices that surround the pore's narrowest segment, presumably its gate. At first consideration a force directed along the triskelion arms toward the center of the trimer, associated with flattening of Piezo's arms, might be expected to constrict the pore further. However, given that TM37-38 are domain-swapped relative to TM1-36, such a force will more likely push the 'swapped' pore-lining helices away from the center and open the pore.

In summary, the structure of Piezo1 leads us to propose a membrane dome mechanism for the origins of its mechanosensitive gating. In this mechanism a dome of membrane, created by Piezo's shape in its closed conformation, undergoes relative flattening upon channel opening. This mechanism does not require the application of a force pressing onto the dome (i.e. a force component normal to the plane of the membrane): lateral membrane tension alone will favor the flatter, opened conformation by a relative energy difference given by $\gamma \Delta A_{proj}$. Whether Piezo interacts directly with cytoskeletal or extracellular matrix proteins to modulate its mechanosensitive gating is unknown (*Lewis and Grandl, 2015*; *Gottlieb et al., 2012*). The membrane dome mechanism presented here is a hypothesis inspired by the highly unusual structure of Piezo and its demonstrated ability to curve lipid bilayers into a dome. Additional experiments will be needed to test this hypothesis.

## Materials and methods

**Key resources table**

| Reagent type (species) or resource | Designation | Source or reference | Identifiers | Additional information |
| --- | --- | --- | --- | --- |
| Gene (*Mus musculus*) | Piezo1 | doi: 10.1126/science.1193270 | UniProt: E2JF22 | |
| Cell line (*Homo sapiens*) | HEK293S GnTI⁻ | ATCC | ATCC: CRL-3022 | |
| Cell line (*Spodoptera frugiperda*) | sf9 | ATCC | ATCC: CRL-1711 | |
| Recombinant DNA reagent | pEG BacMam | doi: 10.1038/nprot.2014.173 | | |
| Software, algorithm | RELION | doi: 10.1016/j.jsb.2012.09.006 | | |
| Software, algorithm | cryoSPARC | doi: 10.1038/nmeth.4169 | | |

## Expression and purification

DNA encoding the full-length mouse Piezo1 protein (UniProt accession: E2JF22) in a pEG BacMam vector was used for expression. A green fluorescent protein (GFP) tag following a PreScission protease cleavage site (LEVLFQ/GP) was placed at the C-terminus to facilitate detection and purification. mPiezo1 was expressed using the BacMam method (*Goehring et al., 2014*). Briefly, baculoviruses carrying mPiezo1 were produced and amplified for two rounds in *Spodoptera frugiperda* Sf9 cells (RRID:CVCL_0549). HEK293S GnTI- cells (RRID:CVCL_A785) in suspension culture were grown at 37°C and infected with 10% (v/v) viruses at a density of $\sim 3 \times 10^6$ cells/ml. At 15 hr post infection, 10 mM sodium butyrate was added to induce expression at 30°C. Cells were harvested at 48 hr post-induction. mPiezo1 purification was kept at 4°C at all times. Cell pellet from 2 L of culture were resuspended in 100 ml TBS supplemented with 1 mM PMSF, 1 µg/ml pepstatin, 1 µg/ml aprotinin, 1 µg/ml leupeptin, 1 mM benzamidine, 1 mM AEBSF, and 0.1 mg/ml trypsin inhibitor. After sonication, the lysate was clarified by centrifugation at $9500 \times g$ for 15 min. The supernatant was then ultra-centrifuged using a Beckman Ti70 rotor at 42,000 rpm (rcf ~181,000 × $g$) at 4°C for 1.5 hr. Membranes were collected and the pellet resuspended and homogenized in 20 ml TBS containing all protease inhibitors as described above. The membrane resuspension was mixed with 20 ml TBS containing 2% digitonin and rotated at 4°C for 1.5 hr for protein extraction. Unsolubilized debris was removed by ultra-centrifugation using a Beckman Ti70 rotor at 42,000 rpm (rcf ~181,000 × $g$) at 4°C for 40 min. The supernatant was incubated with GFP nanobody-coupled Sepharose resin (*Kirchhofer et al., 2010*) equilibrated with elution buffer (150 mM NaCl, 20 mM Tris pH8, 0.05% digitonin, protease inhibitors) for 2 hr. After loading onto a column and collecting the flow-through, the resin was washed with 10 column volumes of wash buffer (300 mM NaCl, 20 mM Tris pH8, 0.05% digitonin, protease inhibitors). PreScission protease was added to the resin at a target protein to protease ratio of 20:1 (w/w) and incubated overnight with gentle rotation to cleave GFP. Target protein was then eluted with elution buffer and concentrated to 500 µl by Amicon Ultra centrifugal filter (MWCO 10 kDa). The sample was injected onto a Superose 6 Increase 10/300 GL column (GE Healthcare, Little Chalfont, United Kingdom) equilibrated with elution buffer. Peak fractions were pooled together and concentrated to 15 mg/ml.

## Cell lines

Cell lines were acquired from and authenticated by the American Type Culture Collection (ATCC). The cell lines were not tested for mycoplasma contamination.

## Electron microscopy sample preparation and imaging

3 mM fluorinated Fos-Choline-8 (FFC-8) was added to freshly purified mPiezo1 sample immediately prior to freezing. Quantifoil 400 mesh gold R1.2/1.3 holey carbon grids were treated with 10 s of glow discharge in reduced pressure air. A first drop of 3 µl protein sample was applied on the grid, incubated for 15 s, and manually removed using a filter paper with minimum touching. Then, a second drop of 3 µl protein sample was added, incubated for 15 s, and then blotted once for 1 s with −1 force and plunged into liquid ethane, using a Vitrobot Mark IV (FEI company, Hillsboro, Oregon) operated at room temperature and 100% humidity. The grids were then stored in liquid nitrogen until imaging.

Automated data collection was controlled by SerialEM (*Mastronarde, 2005*) on a Titan Krios transmission electron microscope (FEI) operating at 300 keV equipped with a K2 Summit direct electron detector (Gatan, Inc., Pleasanton, CA). Micrographs were recorded in super-resolution mode, with a calibrated physical pixel size of 1.3 Å (a super-resolution pixel size of 0.65 Å) and a nominal defocus range of 1.0 to 2.4 µm. The exposure time for each image was 10 s fractionated over 50 frames with a dose rate of 8 electrons per physical pixel per second, corresponding to a total cumulative dose of of 47 electron per $\text{Å}^2$, or 0.94 electron per $\text{Å}^2$ per frame.

## Image processing and map calculation

Whole-frame motion correction was performed with gain reference applied and dose weighting using MotionCor2 (*Zheng et al., 2017*). The contrast transfer function parameters were estimated for all summed images using CTFFIND4 (*Rohou and Grigorieff, 2015*). RELION (*Kimanius et al., 2016*; *Scheres, 2012*) was used for image processing except where noted otherwise. 2000 particles

were manually picked from a subset of images and extracted in a box size of 400 pixels and a mask diameter of 300 Å. Extracted particles were subjected to 2D classification requesting 20 classes, 12 of which showed representative views and were selected as templates for automated particle picking. 576,191 particles picked from 3414 images were then manually inspected to remove false positives, resulting in 521,152 particles. Another round of 2D classification requesting 100 classes was used to further clean up the dataset, which then contained 459,918 particles. Using CryoSPARC (*Punjani et al., 2017*) *ab initio* reconstruction requesting three classes with no symmetry imposed, an initial model was generated from 232,799 particles. This exhibited clear three-fold symmetry. After another round of symmetry-free *ab initio* reconstruction in CryoSPARC to remove non-symmetric particles, the resultant 161,986 particles were reconstructed to 3.74 Å resolution according a 0.143 cutoff criterion on the Fourier shell correlation (FSC) curve, by CryoSPARC homogeneous refinement with C3 symmetry imposed.

Given that the peripheral region of each arm is relatively flexible, the triskelion is not strictly C3-symmetrical. Therefore, symmetry expansion was used to improve the peripheral resolution. The manually inspected 521,152 particles were first aligned in RELION 3D auto-refine with C3 symmetry using the CryoSPARC model as a reference. Using the relion_particle_symmetry_expand program, each particle was then replicated and 120° or 240° was added to the first Euler angle so that each of the three arms of every particle was rotated to the same orientation on the C3-symmetric ring. This 3-fold enlarged dataset was then classified in C1 with a mask covering only a single protruding arm without angular search. The best class containing 277,548 expanded particles was subjected to masked refinement in C1 with the mask covering the central hub and one arm. Only local angular searches were carried out during this refinement so that copies of the same arm do not contribute to the reconstruction more than once. The resultant map has a resolution of 3.8 Å after post-processing and allowed *de novo* model building for the peripheral region. Local resolution was estimated using Blocres (*Cardone et al., 2013*) with a kernel size of 19.

## Model building and refinement

The density map from masked refinement in C1 was sharpened by applying an isotropic b-factor of $-200$ Å$^2$ in RELION post-processing for model building. A crystal structure of the mPiezo1 CED (PDB ID 4RAX) was docked into the map using UCSF Chimera (RRID:SCR_004097) (*Pettersen et al., 2004*) as a starting point. Manual building and refinement was performed in COOT (RRID:SCR_014222) (*Emsley et al., 2010*). Major helix elements were first placed into density and then connected by manually building the loops in baton mode. Sequence was registered by assigning the bulky side chains. The final model contains residues 577–600, 605–717, 782–875, 880–886, 892–1365, 1493–1578, 1655–1807, 1952–1997, 2015–2065, 2075–2411, 2424–2456, 2463–2546. Among them, side chains of residues 577–600, 605–717 and 1551–1578 were trimmed to Cβ, as these densities are less well-defined. In addition, a separate chain containing 16 alanine residues was modeled into density near the central bottom; these are not connected to other parts of the map. A trimer model of the channel was generated from a monomer by applying three-fold symmetry. The trimer model was refined with phenix.real_space_refine (*Afonine et al., 2013*) using secondary structure and non-crystallographic symmetry (NCS) restraints. MolProbity (RRID:SCR_014226) (*Chen et al., 2010*) was used to monitor the model geometry through multiple rounds of real space refinement and manual rebuilding. The refined model has a MolProbity score of 1.73, a clashscore of 4.84, with 0.25% rotamer outliers. The Ramachandran plot contains 91.96% favored, 8.04% allowed, and no outliers.

Structure figures were generated with Chimera, Pymol (RRID:SCR_000305, The PyMOL Molecular Graphics System, Version 1.8 Schrödinger, LLC.), APBS (RRID:SCR_008387) (*Dolinsky et al., 2004*) and HOLE (*Smart et al., 1996*). Structure calculations were performed using the SBGrid suite of programs (*Morin et al., 2013*).

## Proteoliposome reconstitution and imaging

Two different lipid compositions were used for reconstitution. One contains 1-palmitoyl-2-oleoyl-sn-glycero-3-phosphoethanolamine (POPE) and 1-palmitoyl-2-oleoyl-sn-glycero-3-phospho-(1'-rac-glycerol) (POPG) (Avanti Polar Lipids, Alabaster, AL) at a 3:1 wt ratio. The other contains 1-palmitoyl-2-oleoyl-sn-glycero-3-phosphocholine (POPC), 1,2-dioleoyl-sn-glycero-3-phospho-L-serine (DOPS)

(Avanti Polar Lipids) and cholesterol at a 8:1:1 wt ratio. The lipids were mixed in chloroform and washed with pentane. After drying with an argon stream, the lipids were incubated overnight in a vacuum chamber. Dried lipids were suspended by sonication in buffer containing 20 mM Tris pH 8.0, 150 mM NaCl, and then mixed with 1.3% C12E10 for 2 hr at room temperature, to make final lipid concentration 20 mg/ml. Mouse Piezo1 was purified following the same protocol as described above, except replacing digitonin with C12E10, 2% (w/v) and 0.025% (w/v) for extraction and stabilization, respectively. Purified protein was added to the lipid/detergent mixture in a protein-to-lipid ratio of 1:20 (w/w). Following 1.5 hr incubation at 4°C, C12E10 was removed by incubating the mixture with SM-2 bio-beads (Bio-rad, Hercules, CA) overnight at 4°C for the POPE:POPG vesicles, and by dialyzing against detergent-free buffer for 4 days at 4°C for the POPC:DOPS:cholesterol vesicles. Generated proteoliposome vesicles were collected and briefly sonicated before freezing on C-flat 1.2/1.3 400 mesh Holey Carbon grids for the POPE:POPG vesicles, and Quantifoil 400 mesh gold R1.2/1.3 holey carbon grids for the POPC:DOPS:cholesterol vesicles. Micrographs were collected on a Talos Arctica transmission electron microscope (FEI) operating at 200 keV equipped with a K2 Summit direct electron detector (Gatan), in super-resolution mode, with a nominal defocus range of 0.8 to 2.4 μm. The calibrated physical pixel size is 1.5 Å for the POPE:POPG vesicles, and 1.9 Å for the POPC:DOPS:cholesterol vesicles. The exposure time for each image was 10 s fractionated over 50 frames, with a dose rate of 8 electrons per physical pixel per second for the POPE:POPG vesicles, and a dose rate of 15 electrons per physical pixel per second for the POPC:DOPS:cholesterol vesicles. Whole-frame motion correction was performed with gain reference applied and dose weighting using MotionCor2.

## Data availability

Atomic coordinates of mPiezo1 have been deposited in the Protein Data Bank (http://www.rcsb.org) under ID 6B3R. The cryoEM maps (refined in cryoSPARC with C3 symmetry and focused-refined in RELION with C1 symmetry) have been deposited in the Electron Microscopy Data Bank (https://www.ebi.ac.uk/pdbe/emdb) under ID EMD-7042.

## Acknowledgements

We thank A. Patapoutian for the mouse Piezo1 gene, members of the MacKinnon lab for assistance at various stages of the project, C-H Lee for assistance in image processing, X Tao for advice preparing figures, Y-C Hsiung for help with cell culture, M Ebrahim and J Sotiris for support in data collection, G von Heijne, R Phillips, J Chen, K Swartz and S Scheuring for helpful discussions and manuscript review and AB for inspiration. RM is an Investigator in the Howard Hughes Medical Institute.

## Additional information

### Funding

| Funder | Grant reference number | Author |
| --- | --- | --- |
| Howard Hughes Medical Institute | Investigator | Roderick MacKinnon |

The funders had no role in study design, data collection and interpretation, or the decision to submit the work for publication.

### Author contributions

Yusong R Guo, Formal analysis, Validation, Investigation, Visualization, Methodology, Writing—original draft, Writing—review and editing; Roderick MacKinnon, Conceptualization, Resources, Formal analysis, Supervision, Funding acquisition, Methodology, Writing—original draft, Project administration, Writing—review and editing

## Author ORCIDs

Yusong R Guo (iD) http://orcid.org/0000-0002-8563-3397
Roderick MacKinnon (iD) http://orcid.org/0000-0001-7605-4679

## Decision letter and Author response

Decision letter https://doi.org/10.7554/eLife.33660.027
Author response https://doi.org/10.7554/eLife.33660.028

## Additional files

### Supplementary files

• Transparent reporting form
DOI: https://doi.org/10.7554/eLife.33660.020

• Reporting standard 1
DOI: https://doi.org/10.7554/eLife.33660.021

### Major datasets

The following datasets were generated:

| Author(s) | Year | Dataset title | Dataset URL | Database, license, and accessibility information |
|---|---|---|---|---|
| Yusong R Guo, Roderick MacKinnon | 2017 | Structure of the mechanosensitive channel Piezo1 | https://www.rcsb.org/pdb/explore/explore.do?structureId=6B3R | Publicly available at the RCSB Protein Data Bank (accession no: 6B3R) |
| Yusong R Guo, Roderick MacKinnon | 2017 | Structure of the mechanosensitive channel Piezo1 | http://www.ebi.ac.uk/pdbe/entry/emdb/EMD-7042 | Publicly available at the Electron Microscopy Data Bank (accession no: EMD-7042) |

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
