## [Decision Letter]

Thank you for submitting your article "The structure of Piezo reveals a "molecular touch dome" poised for mechanosensation" for consideration by *eLife*. Your article has been reviewed by three peer reviewers, and the evaluation has been overseen by Kenton Swartz as the Reviewing Editor and Richard Aldrich as the Senior Editor. The following individual involved in review of your submission has agreed to reveal his identity: Nikolaus Grigorieff (Reviewer #3).

The reviewers have discussed the reviews with one another and the Reviewing Editor has drafted this decision to help you prepare a revised submission.

Summary:

The authors present a near-atomic-resolution model of the mechanosensitive mouse PIEZO1 channel. While the overall architecture of the channel broadly resembles the trimeric, three-bladed propeller of the 2015 medium-resolution structure published in Nature, this new structure represents a very significant advance and should be an invaluable tool for future structure-function studies.

Strikingly, the three "arms" surrounding the central pore domain point up at a ~30 degree angle making it impossible for them to reside in a planar membrane. To see if this arm orientation is preserved when the channel is embedded in a membrane (rather than the detergent micelles used for the structure determination), the authors image channels reconstituted into small unilamellar vesicles and conclude that in the closed state the channel arms distorts the membrane into a tightly-curved (R = 10nm) dome-shape. They then propose that these arms rotate to a flatter geometry in the open state, as the large (~50 nm^2^) increase in projected membrane area make then make the channel exceptionally sensitive to changes in membrane tension.

Overall, this is an important and fascinating study that will have major impact on the field of mechanosensation, and as such is highly appropriate for publication in *eLife*. The following are points the authors should address in revision.

Essential revisions:

1) The authors present just a single image of a distorted small unilamellar vesicle (Figure 5) to support their hypothesis that the channel reshapes the membrane into a tightly-curved dome shape. However, the shape of vesicles in cryo-EM images is very sensitive to sample preparation (e.g. Methods in Enzymology, Vol 391) and the example in Figure 5 appears as if it could be partially dehydrated. To exclude these effects, the authors could present images of multiple vesicles containing Piezo1 along with control vesicles (i.e. no protein, or with a planar protein like bR) and quantify the curvature of the phospholipid head-group contour in the vicinity of the electron density of the CED. Also, we imagine the authors have already considered opening the channels by intentionally increasing the solution osmolarity (and thus vesicle membrane tension), but perhaps the change in "dome shape" is too small to resolve?

2) The authors' model (Figure 1 and Figure 7) only considers the effect of the average lateral force within the plane of membrane (i.e. tension). While the osmotic-release function of MscL only requires it to sense these lateral forces, there is quite a bit of evidence suggesting that eukaryotic mechanosensitive channels are subjected to vertical forces perpendicular to the plane of membrane via interactions with the cytoskeleton, extra-cellular matrix and even the hydrostatic pressure difference between the cytosol and extra-cellular fluid. Thus, it would be helpful if the authors compared the coupling of their proposed dome flattening/arm rotation to pN-scale vertical forces to the lateral forces they have already considered.

3) The authors made some change to the standard image processing protocol to enhance the resolution in the N-terminal domains forming the somewhat flexible arms of the triskelion. This "symmetry expansion" should be explained in more detail.

4) The authors spend considerable space providing a detailed model for how pushing on the "dome" might open the channel and contrasting this proposed mechanism to gating of MscL. Although elegant, the presentation of the dome model tends to undervalue the physiological niche where these molecules normally reside. It would be helpful for the authors to discuss how factors, such as cytoskeleton, lipid composition, co-factors and/or ECM binding, might be factored into their model. Indeed, there are already several published studies on Piezo channel regulation in the literature that could and should be incorporated or discussed in more detail.

5) It would be helpful to have a diagram highlighting the locations of unresolved intracellular loops, similar to what is shown for the N-terminus. Unlike normal proteins, these intracellular loops can be quite big (over 100 amino acids) and knowing where there still exist gaps in the structure would be useful.

---

## [Author Response]

Summary:The authors present a near-atomic-resolution model of the mechanosensitive mouse PIEZO1 channel. While the overall architecture of the channel broadly resembles the trimeric, three-bladed propeller of the 2015 medium-resolution structure published in Nature, this new structure represents a very significant advance and should be an invaluable tool for future structure-function studies.Strikingly, the three "arms" surrounding the central pore domain point up at a ~30 degree angle making it impossible for them to reside in a planar membrane. To see if this arm orientation is preserved when the channel is embedded in a membrane (rather than the detergent micelles used for the structure determination), the authors image channels reconstituted into small unilamellar vesicles and conclude that in the closed state the channel arms distorts the membrane into a tightly-curved (R = 10nm) dome-shape. They then propose that these arms rotate to a flatter geometry in the open state, as the large (~50 nm^2^) increase in projected membrane area make then make the channel exceptionally sensitive to changes in membrane tension.

We estimate a maximum possible in-plane expansion (ΔA_proj_) of 120 nm^2^. The true value is likely to be smaller as the 120 nm^2^ estimate is an upper limit.

Essential revisions:1) The authors present just a single image of a distorted small unilamellar vesicle (Figure 5) to support their hypothesis that the channel reshapes the membrane into a tightly-curved dome shape. However, the shape of vesicles in cryo-EM images is very sensitive to sample preparation (e.g. Methods in Enzymology, Vol 391) and the example in Figure 5 appears as if it could be partially dehydrated. To exclude these effects, the authors could present images of multiple vesicles containing Piezo1 along with control vesicles (i.e. no protein, or with a planar protein like bR) and quantify the curvature of the phospholipid head-group contour in the vicinity of the electron density of the CED. Also, we imagine the authors have already considered opening the channels by intentionally increasing the solution osmolarity (and thus vesicle membrane tension), but perhaps the change in "dome shape" is too small to resolve?

We have addressed these concerns by adding images of vesicles consisting of a different lipid composition in which vesicles are spherical in the absence of Piezo (Figure 6) and show local deformation in the presence of Piezo (Figure 6 and Figure 6—figure supplement 1). Images with dashed circles are included to highlight the local deviations from the global curvature of the vesicles. We provide additional text to present these data in the section entitled “Membrane curving properties of Piezo”.

We think the non-spherical shape of vesicles made of POPE and POPG most likely reflects the chemical nature and shape of these particular lipids, which can favor membranes with intrinsic curvature. This raises the interesting possibility that in cells the lipid composition very near Piezo channels might reflect the curvature that this channel imposes on the membrane. As to the important point raised by this referee, the POPC-DOPS-cholesterol containing vesicles are spherical and Piezo still promotes a local dome of deformation. And yes, of course we are carrying out experiments to assess the channel/dome conformation as a function of osmotic pressure, which in principle is translatable to tension through radius of curvature and the Young-Laplace Equation. More to follow…

2) The authors' model (Figure 1 and Figure 7) only considers the effect of the average lateral force within the plane of membrane (i.e. tension). While the osmotic-release function of MscL only requires it to sense these lateral forces, there is quite a bit of evidence suggesting that eukaryotic mechanosensitive channels are subjected to vertical forces perpendicular to the plane of membrane via interactions with the cytoskeleton, extra-cellular matrix and even the hydrostatic pressure difference between the cytosol and extra-cellular fluid. Thus, it would be helpful if the authors compared the coupling of their proposed dome flattening/arm rotation to pN-scale vertical forces to the lateral forces they have already considered.

In the case of Piezo it appears that lateral membrane tension alone can mediate gating (references made in the Introduction). Moreover, the evidence for a role of DIRECT interactions with cytoskeletal or extracellular matrix proteins to gate the channel is not very strong (also referenced in the introduction). (Piezo clearly shows mechanosensitive gating in the absence of intact cytoskeletal elements, so it is likely that whatever influence other proteins may have, they are likely to be acting indirectly.)

Thus, we only consider the average lateral force because, based on this unusual structure and its ability to curve the membrane into a dome, we can explain mechanosensitivity on the basis of it alone. We do not exclude the possible existence of other forces with components normal to the membrane plane. However, because we are describing a new possible mechanism for mechanosensation, we wish to focus on it and explain it clearly. We reiterate this point in the revised, final paragraph.

3) The authors made some change to the standard image processing protocol to enhance the resolution in the N-terminal domains forming the somewhat flexible arms of the triskelion. This "symmetry expansion" should be explained in more detail.

We have added a more complete description of the symmetry expansion in the section entitled “Image processing and map calculation” in the Materials and methods section.

4) The authors spend considerable space providing a detailed model for how pushing on the "dome" might open the channel and contrasting this proposed mechanism to gating of MscL. Although elegant, the presentation of the dome model tends to undervalue the physiological niche where these molecules normally reside. It would be helpful for the authors to discuss how factors, such as cytoskeleton, lipid composition, co-factors and/or ECM binding, might be factored into their model. Indeed, there are already several published studies on Piezo channel regulation in the literature that could and should be incorporated or discussed in more detail.

The model does not describe how pushing on the dome might open the channel – that’s not how it works. We failed to convey the mechanism properly and have revised the presentation in several places throughout the manuscript. These revisions include a change of title, realizing that “molecular touch dome” may be misleading. The revisions also include a careful description of the concept that out of plane membrane (the dome) is transferred into plane to lower the free energy of the system in a membrane under tension. Thus, in the mechanism, the hypothesized flatter, open conformation is favored in a membrane under tension. This transition would be mediated by membrane tension, which would likely be imposed by forces applied elsewhere on the membrane (not by pushing on the dome.)

Regarding the physiological niche, it seems possible that different cells by virtue of their shape, attachments to other surfaces, membrane lipid composition (see response to point 1) and other factors can create more or less favorable environments for the function of a channel that operates by the mechanism we are putting forth. To invent an example, a cell with lots of excess, untethered surface area would seem to leave membrane dome mechanism channels permanently closed. A cell in which the membrane (or parts of it) are somehow held on the edge of zero tension would seem better poised for activation of membrane dome mediated channels. The problem is, at present we do not have enough information about such niches. Our goal is to propose a principle by which this channel might work. That’s all we are trying to do here.

We already referenced papers trying to address the role of cytoskeleton and stated our opinion of them.

5) It would be helpful to have a diagram highlighting the locations of unresolved intracellular loops, similar to what is shown for the N-terminus. Unlike normal proteins, these intracellular loops can be quite big (over 100 amino acids) and knowing where there still exist gaps in the structure would be useful.

We added dotted lines to Figure 3 that are in approximate length proportional to the number of missing amino acids. These missing amino acids are listed explicitly in Figure 3—figure supplement 1–Figure 3—figure supplement 3.